# QUASI-RECURRENT NEURAL NETWORKS

**James Bradbury**,* **Stephen Merity**,* **Caiming Xiong & Richard Socher**
Salesforce Research
Palo Alto, California
`{james.bradbury,smerity,cxiong,rsocher}@salesforce.com`

## ABSTRACT

Recurrent neural networks are a powerful tool for modeling sequential data, but the dependence of each timestep's computation on the previous timestep's output limits parallelism and makes RNNs unwieldy for very long sequences. We introduce quasi-recurrent neural networks (QRNNs), an approach to neural sequence modeling that alternates convolutional layers, which apply in parallel across timesteps, and a minimalist recurrent pooling function that applies in parallel across channels. Despite lacking trainable recurrent layers, stacked QRNNs have better predictive accuracy than stacked LSTMs of the same hidden size. Due to their increased parallelism, they are up to 16 times faster at train and test time. Experiments on language modeling, sentiment classification, and character-level neural machine translation demonstrate these advantages and underline the viability of QRNNs as a basic building block for a variety of sequence tasks.

## 1 INTRODUCTION

Recurrent neural networks (RNNs), including gated variants such as the long short-term memory (LSTM) (Hochreiter & Schmidhuber, 1997) have become the standard model architecture for deep learning approaches to sequence modeling tasks. RNNs repeatedly apply a function with trainable parameters to a hidden state. Recurrent layers can also be stacked, increasing network depth, representational power and often accuracy. RNN applications in the natural language domain range from sentence classification (Wang et al., 2015) to word- and character-level language modeling (Zaremba et al., 2014). RNNs are also commonly the basic building block for more complex models for tasks such as machine translation (Bahdanau et al., 2015; Luong et al., 2015; Bradbury & Socher, 2016) or question answering (Kumar et al., 2016; Xiong et al., 2016). Unfortunately standard RNNs, including LSTMs, are limited in their capability to handle tasks involving very long sequences, such as document classification or character-level machine translation, as the computation of features or states for different parts of the document cannot occur in parallel.

Convolutional neural networks (CNNs) (Krizhevsky et al., 2012), though more popular on tasks involving image data, have also been applied to sequence encoding tasks (Zhang et al., 2015). Such models apply time-invariant filter functions in parallel to windows along the input sequence. CNNs possess several advantages over recurrent models, including increased parallelism and better scaling to long sequences such as those often seen with character-level language data. Convolutional models for sequence processing have been more successful when combined with RNN layers in a hybrid architecture (Lee et al., 2016), because traditional max- and average-pooling approaches to combining convolutional features across timesteps assume time invariance and hence cannot make full use of large-scale sequence order information.

We present quasi-recurrent neural networks for neural sequence modeling. QRNNs address both drawbacks of standard models: like CNNs, QRNNs allow for parallel computation across both timestep and minibatch dimensions, enabling high throughput and good scaling to long sequences. Like RNNs, QRNNs allow the output to depend on the overall order of elements in the sequence. We describe QRNN variants tailored to several natural language tasks, including document-level sentiment classification, language modeling, and character-level machine translation. These models outperform strong LSTM baselines on all three tasks while dramatically reducing computation time.

---

*Equal contribution



Figure 1: Block diagrams showing the computation structure of the QRNN compared with typical LSTM and CNN architectures. Red signifies convolutions or matrix multiplications; a continuous block means that those computations can proceed in parallel. Blue signifies parameterless functions that operate in parallel along the channel/feature dimension. LSTMs can be factored into (red) linear blocks and (blue) elementwise blocks, but computation at each timestep still depends on the results from the previous timestep.

## 2 MODEL

Each layer of a quasi-recurrent neural network consists of two kinds of subcomponents, analogous to convolution and pooling layers in CNNs. The convolutional component, like convolutional layers in CNNs, allows fully parallel computation across both minibatches and spatial dimensions, in this case the sequence dimension. The pooling component, like pooling layers in CNNs, lacks trainable parameters and allows fully parallel computation across minibatch and feature dimensions.

Given an input sequence $\mathbf{X} \in \mathbb{R}^{T \times n}$ of $T$ $n$-dimensional vectors $\mathbf{x}_1 \ldots \mathbf{x}_T$, the convolutional sub-component of a QRNN performs convolutions in the timestep dimension with a bank of $m$ filters, producing a sequence $\mathbf{Z} \in \mathbb{R}^{T \times m}$ of $m$-dimensional candidate vectors $\mathbf{z}_t$. In order to be useful for tasks that include prediction of the next token, the filters must not allow the computation for any given timestep to access information from future timesteps. That is, with filters of width $k$, each $\mathbf{z}_t$ depends only on $\mathbf{x}_{t-k+1}$ through $\mathbf{x}_t$. This concept, known as a masked convolution (van den Oord et al., 2016a), is implemented by padding the input to the left by the convolution's filter size minus one.

We apply additional convolutions with separate filter banks to obtain sequences of vectors for the elementwise gates that are needed for the pooling function. While the candidate vectors are passed through a $\tanh$ nonlinearity, the gates use an elementwise sigmoid. If the pooling function requires a forget gate $\mathbf{f}_t$ and an output gate $\mathbf{o}_t$ at each timestep, the full set of computations in the convolutional component is then:

$$
\begin{aligned}
\mathbf{Z} &= \tanh(\mathbf{W}_z * \mathbf{X}) \\
\mathbf{F} &= \sigma(\mathbf{W}_f * \mathbf{X}) \\
\mathbf{O} &= \sigma(\mathbf{W}_o * \mathbf{X}),
\end{aligned}
\tag{1}
$$

where $\mathbf{W}_z, \mathbf{W}_f$, and $\mathbf{W}_o$, each in $\mathbb{R}^{k \times n \times m}$, are the convolutional filter banks and $*$ denotes a masked convolution along the timestep dimension. Note that if the filter width is 2, these equations reduce to the LSTM-like

$$
\begin{aligned}
\mathbf{z}_t &= \tanh(\mathbf{W}_z^1 \mathbf{x}_{t-1} + \mathbf{W}_z^2 \mathbf{x}_t) \\
\mathbf{f}_t &= \sigma(\mathbf{W}_f^1 \mathbf{x}_{t-1} + \mathbf{W}_f^2 \mathbf{x}_t) \\
\mathbf{o}_t &= \sigma(\mathbf{W}_o^1 \mathbf{x}_{t-1} + \mathbf{W}_o^2 \mathbf{x}_t).
\end{aligned}
\tag{2}
$$

Convolution filters of larger width effectively compute higher $n$-gram features at each timestep; thus larger widths are especially important for character-level tasks.

Suitable functions for the pooling subcomponent can be constructed from the familiar elementwise gates of the traditional LSTM cell. We seek a function controlled by gates that can mix states across timesteps, but which acts independently on each channel of the state vector. The simplest option, which Balduzzi & Ghifary (2016) term "dynamic average pooling", uses only a forget gate:

$$
\mathbf{h}_t = \mathbf{f}_t \odot \mathbf{h}_{t-1} + (1 - \mathbf{f}_t) \odot \mathbf{z}_t,
\tag{3}
$$

where $\odot$ denotes elementwise multiplication. The function may also include an output gate:

$$\begin{aligned}
\mathbf{c}_t &= \mathbf{f}_t \odot \mathbf{c}_{t-1} + (1 - \mathbf{f}_t) \odot \mathbf{z}_t \\
\mathbf{h}_t &= \mathbf{o}_t \odot \mathbf{c}_t.
\end{aligned} \tag{4}$$

Or the recurrence relation may include an independent input and forget gate:

$$\begin{aligned}
\mathbf{c}_t &= \mathbf{f}_t \odot \mathbf{c}_{t-1} + \mathbf{i}_t \odot \mathbf{z}_t \\
\mathbf{h}_t &= \mathbf{o}_t \odot \mathbf{c}_t.
\end{aligned} \tag{5}$$

We term these three options $f$-pooling, $fo$-pooling, and $ifo$-pooling respectively; in each case we initialize $\mathbf{h}$ or $\mathbf{c}$ to zero. Although the recurrent parts of these functions must be calculated for each timestep in sequence, their simplicity and parallelism along feature dimensions means that, in practice, evaluating them over even long sequences requires a negligible amount of computation time.

A single QRNN layer thus performs an input-dependent pooling, followed by a gated linear combination of convolutional features. As with convolutional neural networks, two or more QRNN layers should be stacked to create a model with the capacity to approximate more complex functions.

## 2.1 Variants

Motivated by several common natural language tasks, and the long history of work on related architectures, we introduce several extensions to the stacked QRNN described above. Notably, many extensions to both recurrent and convolutional models can be applied directly to the QRNN as it combines elements of both model types.

**Regularization** An important extension to the stacked QRNN is a robust regularization scheme inspired by recent work in regularizing LSTMs.

The need for an effective regularization method for LSTMs, and dropout's relative lack of efficacy when applied to recurrent connections, led to the development of recurrent dropout schemes, including variational inference–based dropout (Gal & Ghahramani, 2016) and zoneout (Krueger et al., 2016). These schemes extend dropout to the recurrent setting by taking advantage of the repeating structure of recurrent networks, providing more powerful and less destructive regularization.

Variational inference–based dropout locks the dropout mask used for the recurrent connections across timesteps, so a single RNN pass uses a single stochastic subset of the recurrent weights. Zoneout stochastically chooses a new subset of channels to "zone out" at each timestep; for these channels the network copies states from one timestep to the next without modification.

As QRNNs lack recurrent weights, the variational inference approach does not apply. Thus we extended zoneout to the QRNN architecture by modifying the pooling function to keep the previous pooling state for a stochastic subset of channels. Conveniently, this is equivalent to stochastically setting a subset of the QRNN's $f$ gate channels to 1, or applying dropout on $1 - f$:

$$\mathbf{F} = 1 - \text{dropout}(1 - \sigma(\mathbf{W}_f * \mathbf{X}))) \tag{6}$$

Thus the pooling function itself need not be modified at all. We note that when using an off-the-shelf dropout layer in this context, it is important to remove automatic rescaling functionality from the implementation if it is present. In many experiments, we also apply ordinary dropout between layers, including between word embeddings and the first QRNN layer.

**Densely-Connected Layers** We can also extend the QRNN architecture using techniques introduced for convolutional networks. For sequence classification tasks, we found it helpful to use skip-connections between every QRNN layer, a technique termed "dense convolution" by Huang et al. (2016). Where traditional feed-forward or convolutional networks have connections only between subsequent layers, a "DenseNet" with $L$ layers has feed-forward or convolutional connections between every pair of layers, for a total of $L(L-1)$. This can improve gradient flow and convergence properties, especially in deeper networks, although it requires a parameter count that is quadratic in the number of layers.

When applying this technique to the QRNN, we include connections between the input embeddings and every QRNN layer and between every pair of QRNN layers. This is equivalent to concatenating

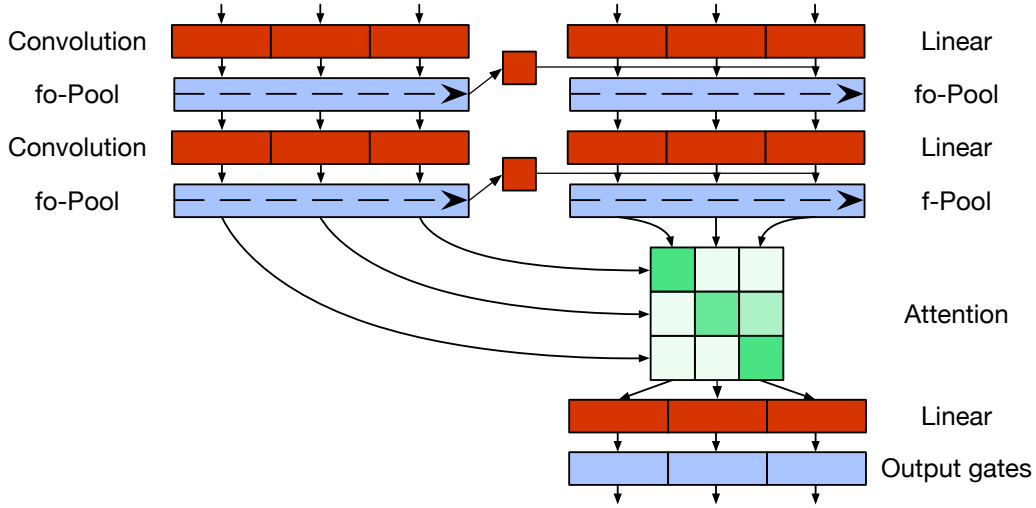

Figure 2: The QRNN encoder–decoder architecture used for machine translation experiments.

each QRNN layer's input to its output along the channel dimension before feeding the state into the next layer. The output of the last layer alone is then used as the overall encoding result.

**Encoder–Decoder Models** To demonstrate the generality of QRNNs, we extend the model architecture to sequence-to-sequence tasks, such as machine translation, by using a QRNN as encoder and a modified QRNN, enhanced with attention, as decoder. The motivation for modifying the decoder is that simply feeding the last encoder hidden state (the output of the encoder's pooling layer) into the decoder's recurrent pooling layer, analogously to conventional recurrent encoder–decoder architectures, would not allow the encoder state to affect the gate or update values that are provided to the decoder's pooling layer. This would substantially limit the representational power of the decoder.

Instead, the output of each decoder QRNN layer's convolution functions is supplemented at every timestep with the final encoder hidden state. This is accomplished by adding the result of the convolution for layer $\ell$ (e.g., $\mathbf{W}_z^\ell * \mathbf{X}^\ell$, in $\mathbb{R}^{T \times m}$) with broadcasting to a linearly projected copy of layer $\ell$'s last encoder state (e.g., $\mathbf{V}_z^\ell \tilde{\mathbf{h}}_T^\ell$, in $\mathbb{R}^m$):

$$
\begin{aligned}
\mathbf{Z}^\ell &= \tanh(\mathbf{W}_z^\ell * \mathbf{X}^\ell + \mathbf{V}_z^\ell \tilde{\mathbf{h}}_T^\ell) \\
\mathbf{F}^\ell &= \sigma(\mathbf{W}_f^\ell * \mathbf{X}^\ell + \mathbf{V}_f^\ell \tilde{\mathbf{h}}_T^\ell) \\
\mathbf{O}^\ell &= \sigma(\mathbf{W}_o^\ell * \mathbf{X}^\ell + \mathbf{V}_o^\ell \tilde{\mathbf{h}}_T^\ell),
\end{aligned}
\tag{7}
$$

where the tilde denotes that $\tilde{\mathbf{h}}$ is an encoder variable. Encoder–decoder models which operate on long sequences are made significantly more powerful with the addition of soft attention (Bahdanau et al., 2015), which removes the need for the entire input representation to fit into a fixed-length encoding vector. In our experiments, we computed an attentional sum of the encoder's last layer's hidden states. We used the dot products of these encoder hidden states with the decoder's last layer's un-gated hidden states, applying a $\operatorname{softmax}$ along the encoder timesteps, to weight the encoder states into an attentional sum $\mathbf{k}_t$ for each decoder timestep. This context, and the decoder state, are then fed into a linear layer followed by the output gate:

$$
\begin{aligned}
\alpha_{st} &= \operatorname*{softmax}_{\text{all } s}(\mathbf{c}_t^L \cdot \tilde{\mathbf{h}}_s^L) \\
\mathbf{k}_t &= \sum_s \alpha_{st} \tilde{\mathbf{h}}_s^L \\
\mathbf{h}_t^L &= \mathbf{o}_t \odot (\mathbf{W}_k \mathbf{k}_t + \mathbf{W}_c \mathbf{c}_t^L),
\end{aligned}
\tag{8}
$$

where $L$ is the last layer. This procedure is closely analogous to the attention mechanism described by Luong et al. (2015) as "global-dot without input feeding". The reason for avoiding input feeding is to allow the QRNN layers to run in a maximally timestep-parallel way during training, even

| Model | Time / Epoch (s) | Test Acc (%) |
|---|---|---|
| NBSVM-bi (Wang & Manning, 2012) | – | 91.2 |
| 2 layer sequential BoW CNN (Johnson & Zhang, 2014) | – | 92.3 |
| Ensemble of RNNs and NB-SVM (Mesnil et al., 2014) | – | 92.6 |
| 2-layer LSTM (Longpre et al., 2016) | – | 87.6 |
| Residual 2-layer bi-LSTM (Longpre et al., 2016) | – | 90.1 |
| *Our models* | | |
| Densely-connected 4-layer LSTM (cuDNN optimized) | 480 | 90.9 |
| Densely-connected 4-layer QRNN | 150 | 91.4 |
| Densely-connected 4-layer QRNN with $k = 4$ | 160 | 91.1 |

Table 1: Accuracy comparison on the IMDb binary sentiment classification task. All of our models use 256 units per layer; all layers other than the first layer, whose filter width may vary, use filter width $k = 2$. Train times are reported on a single NVIDIA K40 GPU. We exclude semi-supervised models that conduct additional training on the unlabeled portion of the dataset.

if they can't during inference; any kind of input feeding would make this impossible, although it would likely result in slightly better translation performance.

While the first step of this attention procedure is quadratic in the sequence length, in practice it takes significantly less computation time than the model's linear and convolutional layers due to the simple and highly parallel dot-product scoring function.

## 3 EXPERIMENTS

We evaluate the performance of the QRNN on three different natural language tasks: document-level sentiment classification, language modeling, and character-based neural machine translation. Our QRNN models outperform LSTM-based models of equal hidden size on all three tasks while dramatically improving computation speed. Experiments were implemented in Chainer (Tokui et al.).

### 3.1 SENTIMENT CLASSIFICATION

We evaluate the QRNN architecture on a popular document-level sentiment classification benchmark, the IMDb movie review dataset (Maas et al., 2011). The dataset consists of a balanced sample of 25,000 positive and 25,000 negative reviews, divided into equal-size train and test sets, with an average document length of 231 words (Wang & Manning, 2012). We compare only to other results that do not make use of additional unlabeled data (thus excluding e.g., Miyato et al. (2016)).

Our best performance on a held-out development set was achieved using a four-layer densely-connected QRNN with 256 units per layer and word vectors initialized using 300-dimensional cased GloVe embeddings (Pennington et al., 2014). Dropout of 0.3 was applied between layers, and we used $L^2$ regularization of $4 \times 10^{-6}$. Optimization was performed on minibatches of 24 examples using RMSprop (Tieleman & Hinton, 2012) with learning rate of 0.001, $\alpha = 0.9$, and $\epsilon = 10^{-8}$.

Small batch sizes and long sequence lengths provide an ideal situation for demonstrating the QRNN's performance advantages over traditional recurrent architectures. This is because traditional architectures are only fully parallel over the batch dimension, while the QRNN parallelizes over batch and timestep dimensions in the convolutional layer and over batch and feature dimensions in the pooling layer. We observed a speedup of 3.2x on IMDb train time per epoch compared to the optimized LSTM implementation provided in NVIDIA's cuDNN library. For specific batch sizes and sequence lengths, a 16x speed gain is possible. Figure 4 provides extensive speed comparisons.

In Figure 3, we visualize the hidden state vectors $\mathbf{c}_t^L$ of the final QRNN layer on part of an example from the IMDb dataset. Even without any post-processing, changes in the hidden state are visible and interpretable in regards to the input. This is a consequence of the elementwise nature of the recurrent pooling function, which delays direct interaction between different channels of the hidden state until the computation of the next QRNN layer.

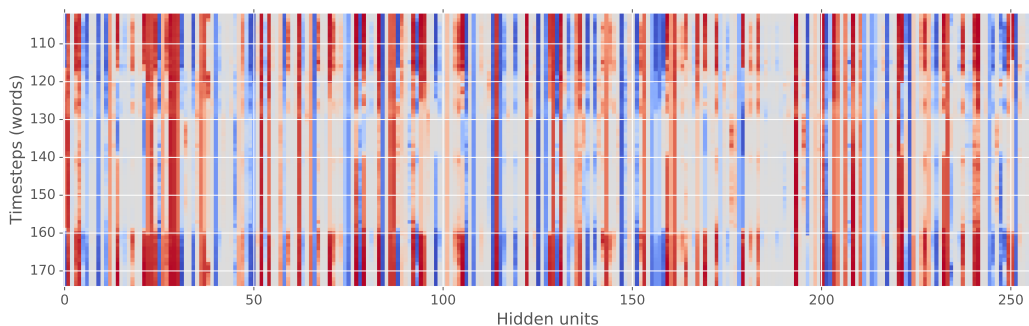

Figure 3: Visualization of the final QRNN layer's hidden state vectors $\mathbf{c}_t^L$ in the IMDb task, with timesteps along the vertical axis. Colors denote neuron activations. After an initial positive statement "*This movie is simply gorgeous*" (off graph at timestep 9), timestep 117 triggers a reset of most hidden states due to the phrase "*not exactly a bad story*" (soon after "*main weakness is its story*"). Only at timestep 158, after "*I recommend this movie to everyone, even if you've never played the game*", do the hidden units recover.

## 3.2 LANGUAGE MODELING

We replicate the language modeling experiment of Zaremba et al. (2014) and Gal & Ghahramani (2016) to benchmark the QRNN architecture for natural language sequence prediction. The experiment uses a standard preprocessed version of the Penn Treebank (PTB) by Mikolov et al. (2010).

We implemented a gated QRNN model with medium hidden size: 2 layers with 640 units in each layer. Both QRNN layers use a convolutional filter width $k$ of two timesteps. While the "medium" models used in other work (Zaremba et al., 2014; Gal & Ghahramani, 2016) consist of 650 units in each layer, it was more computationally convenient to use a multiple of 32. As the Penn Treebank is a relatively small dataset, preventing overfitting is of considerable importance and a major focus of recent research. It is not obvious in advance which of the many RNN regularization schemes would perform well when applied to the QRNN. Our tests showed encouraging results from zoneout applied to the QRNN's recurrent pooling layer, implemented as described in Section 2.1.

The experimental settings largely followed the "medium" setup of Zaremba et al. (2014). Optimization was performed by stochastic gradient descent (SGD) without momentum. The learning rate was set at 1 for six epochs, then decayed by 0.95 for each subsequent epoch, for a total of 72 epochs. We additionally used $L^2$ regularization of $2 \times 10^{-4}$ and rescaled gradients with norm above 10. Zoneout was applied by performing dropout with ratio 0.1 on the forget gates of the QRNN, without rescaling the output of the dropout function. Batches consist of 20 examples, each 105 timesteps.

| Model | Parameters | Validation | Test |
|---|---|---|---|
| LSTM (medium) (Zaremba et al., 2014) | 20M | 86.2 | 82.7 |
| Variational LSTM (medium, MC) (Gal & Ghahramani, 2016) | 20M | 81.9 | 79.7 |
| LSTM with CharCNN embeddings (Kim et al., 2016) | 19M | − | 78.9 |
| Zoneout + Variational LSTM (medium) (Merity et al., 2016) | 20M | 84.4 | 80.6 |
| *Our models* | | | |
| LSTM (medium) | 20M | 85.7 | 82.0 |
| QRNN (medium) | 18M | 82.9 | 79.9 |
| QRNN + zoneout ($p = 0.1$) (medium) | 18M | 82.1 | 78.3 |

Table 2: Single model perplexity on validation and test sets for the Penn Treebank language modeling task. Lower is better. "Medium" refers to a two-layer network with 640 or 650 hidden units per layer. All QRNN models include dropout of 0.5 on embeddings and between layers. MC refers to Monte Carlo dropout averaging at test time.

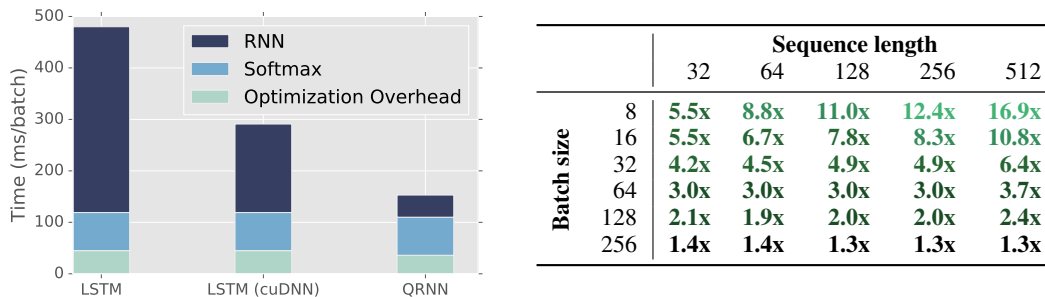

| | | Sequence length | | | | |
|---|---|---|---|---|---|---|
| | | **32** | **64** | **128** | **256** | **512** |
| **Batch size** | 8 | 5.5x | 8.8x | 11.0x | 12.4x | 16.9x |
| | 16 | 5.5x | 6.7x | 7.8x | 8.3x | 10.8x |
| | 32 | 4.2x | 4.5x | 4.9x | 4.9x | 6.4x |
| | 64 | 3.0x | 3.0x | 3.0x | 3.0x | 3.7x |
| | 128 | 2.1x | 1.9x | 2.0x | 2.0x | 2.4x |
| | 256 | 1.4x | 1.4x | 1.3x | 1.3x | 1.3x |

Figure 4: *Left:* Training speed for two-layer 640-unit PTB LM on a batch of 20 examples of 105 timesteps. "RNN" and "softmax" include the forward and backward times, while "optimization overhead" includes gradient clipping, $L^2$ regularization, and SGD computations.
*Right:* Inference speed advantage of a 320-unit QRNN layer alone over an equal-sized cuDNN LSTM layer for data with the given batch size and sequence length. Training results are similar.

Comparing our results on the gated QRNN with zoneout to the results of LSTMs with both ordinary and variational dropout in Table 2, we see that the QRNN is highly competitive. The QRNN without zoneout strongly outperforms both our medium LSTM and the medium LSTM of Zaremba et al. (2014) which do not use recurrent dropout and is even competitive with variational LSTMs. This may be due to the limited computational capacity that the QRNN's pooling layer has relative to the LSTM's recurrent weights, providing structural regularization over the recurrence.

Without zoneout, early stopping based upon validation loss was required as the QRNN would begin overfitting. By applying a small amount of zoneout ($p = 0.1$), no early stopping is required and the QRNN achieves competitive levels of perplexity to the variational LSTM of Gal & Ghahramani (2016), which had variational inference based dropout of 0.2 applied recurrently. Their best performing variation also used Monte Carlo (MC) dropout averaging at test time of 1000 different masks, making it computationally more expensive to run.

When training on the PTB dataset with an NVIDIA K40 GPU, we found that the QRNN is substantially faster than a standard LSTM, even when comparing against the optimized cuDNN LSTM. In Figure 4 we provide a breakdown of the time taken for Chainer's default LSTM, the cuDNN LSTM, and QRNN to perform a full forward and backward pass on a single batch during training of the RNN LM on PTB. For both LSTM implementations, running time was dominated by the RNN computations, even with the highly optimized cuDNN implementation. For the QRNN implementation, however, the "RNN" layers are no longer the bottleneck. Indeed, there are diminishing returns from further optimization of the QRNN itself as the softmax and optimization overhead take equal or greater time. Note that the softmax, over a vocabulary size of only 10,000 words, is relatively small; for tasks with larger vocabularies, the softmax would likely dominate computation time.

It is also important to note that the cuDNN library's RNN primitives do not natively support any form of recurrent dropout. That is, running an LSTM that uses a state-of-the-art regularization scheme at cuDNN-like speeds would likely require an entirely custom kernel.

### 3.3 CHARACTER-LEVEL NEURAL MACHINE TRANSLATION

We evaluate the sequence-to-sequence QRNN architecture described in 2.1 on a challenging neural machine translation task, IWSLT German–English spoken-domain translation, applying fully character-level segmentation. This dataset consists of 209,772 sentence pairs of parallel training data from transcribed TED and TEDx presentations, with a mean sentence length of 103 characters for German and 93 for English. We remove training sentences with more than 300 characters in English or German, and use a unified vocabulary of 187 Unicode code points.

Our best performance on a development set (TED.tst2013) was achieved using a four-layer encoder–decoder QRNN with 320 units per layer, no dropout or $L^2$ regularization, and gradient rescaling to a maximum magnitude of 5. Inputs were supplied to the encoder reversed, while the encoder convolutions were not masked. The first encoder layer used convolutional filter width $k = 6$, while

| Model | Train Time | BLEU (TED.tst2014) |
|---|---|---|
| Word-level LSTM w/attn (Ranzato et al., 2016) | – | 20.2 |
| Word-level CNN w/attn, input feeding (Wiseman & Rush, 2016) | – | 24.0 |
| Char-level ByteNet[1] | – | 24.7 |
| *Our models* | | |
| Char-level 4-layer LSTM | 4.2 hrs/epoch | 16.53 |
| Char-level 4-layer QRNN with $k = 6$ | 1.0 hrs/epoch | 19.41 |

Table 3: Translation performance, measured by BLEU, and train speed in hours per epoch, for the IWSLT German-English spoken language translation task. All models were trained on in-domain data only, and use negative log-likelihood as the training criterion. Our models were trained for 10 epochs. The QRNN model uses $k = 2$ for all layers other than the first encoder layer.

the other encoder layers used $k = 2$. Optimization was performed for 10 epochs on minibatches of 16 examples using Adam (Kingma & Ba, 2014) with $\alpha = 0.001$, $\beta_1 = 0.9$, $\beta_2 = 0.999$, and $\epsilon = 10^{-8}$. Decoding was performed using beam search with beam width 8 and length normalization $\alpha = 0.6$. The modified log-probability ranking criterion is provided in the appendix.

Results using this architecture were compared to an equal-sized four-layer encoder–decoder LSTM with attention, applying dropout of 0.2. We again optimized using Adam; other hyperparameters were equal to their values for the QRNN and the same beam search procedure was applied. Table 3 shows that the QRNN outperformed the character-level LSTM, almost matching the performance of a word-level attentional baseline.

## 4 RELATED WORK

Exploring alternatives to traditional RNNs for sequence tasks is a major area of current research. Quasi-recurrent neural networks are related to several such recently described models, especially the strongly-typed recurrent neural networks (T-RNN) introduced by Balduzzi & Ghifary (2016). While the motivation and constraints described in that work are different, Balduzzi & Ghifary (2016)'s concepts of "learnware" and "firmware" parallel our discussion of convolution-like and pooling-like subcomponents. As the use of a fully connected layer for recurrent connections violates the constraint of "strong typing", all strongly-typed RNN architectures (including the T-RNN, T-GRU, and T-LSTM) are also quasi-recurrent. However, some QRNN models (including those with attention or skip-connections) are not "strongly typed". In particular, a T-RNN differs from a QRNN as described in this paper with filter size 1 and $f$-pooling only in the absence of an activation function on $\mathbf{z}$. Similarly, T-GRUs and T-LSTMs differ from QRNNs with filter size 2 and *fo*- or *ifo*-pooling respectively in that they lack $\tanh$ on $\mathbf{z}$ and use $\tanh$ rather than sigmoid on $\mathbf{o}$.

Another related sequence model is the query-reduction network introduced by Seo et al. (2016). Such a network without its query component could be rewritten as a QRNN with filter size 1, while the full QRN is similar to a single layer of the decoder component of our sequence-to-sequence architecture.

The PixelCNN model (van den Oord et al., 2016a) was the first to tackle a sequence prediction problem (in particular, the computer vision equivalent of language modeling) using masked convolutions in place of recurrent units. Like the QRNN, the PixelCNN architecture allows for highly parallel computation whenever the whole input is available ahead of time (e.g., during training). In order to enable conditional image generation (a setting similar to the QRNN encoder–decoder), the outputs of the convolutions in a PixelCNN can be augmented with a term that depends on the encoder state, while better generation performance was obtained by adding an elementwise gate to the model output (van den Oord et al., 2016b). The PixelCNN, however, relies on depth and large filter sizes to provide long-term context dependence; unlike the QRNN, the gating mechanism is not recurrent.

---

[1] Unpublished result from NIPS 2016 tutorial by Nal Kalchbrenner, given after the submission of this paper (slides at https://drive.google.com/file/d/0B7jhGCaUwDJeZWZWUXJ4cktxVU0/view). See Related Work for discussion.

The QRNN encoder–decoder model shares the favorable parallelism and path-length properties exhibited by the ByteNet (Kalchbrenner et al., 2016), a PixelCNN-like architecture for character-level machine translation based on residual convolutions over binary trees. Their model was constructed to achieve three desired properties: parallelism, linear-time computational complexity, and short paths between any pair of words in order to better propagate gradient signals. While the ByteNet outperforms the QRNN encoder–decoder by about five BLEU points on the IWSLT dataset, it is unclear how much of this difference can be attributed to the overall ByteNet model architecture, as opposed to the many other contributions of that paper, like residual multiplicative blocks or sub-batch normalization.

The QRNN is also related to work in hybrid convolutional–recurrent models. Zhou et al. (2015) apply CNNs at the word level to generate $n$-gram features used by an LSTM for text classification. Xiao & Cho (2016) also tackle text classification by applying convolutions at the character level, with a stride to reduce sequence length, then feeding these features into a bidirectional LSTM. A similar approach was taken by Lee et al. (2016) for character-level machine translation. Their model's encoder uses a convolutional layer followed by max-pooling to reduce sequence length, a four-layer highway network, and a bidirectional GRU. The parallelism of the convolutional, pooling, and highway layers allows training speed comparable to subword-level models without hard-coded text segmentation.

## 5 CONCLUSION

Intuitively, many aspects of the semantics of long sequences are context-invariant and can be computed in parallel (e.g., convolutionally), but some aspects require long-distance context and must be computed recurrently. Many existing neural network architectures either fail to take advantage of the contextual information or fail to take advantage of the parallelism. QRNNs exploit both parallelism and context, exhibiting advantages from both convolutional and recurrent neural networks. QRNNs have better predictive accuracy than LSTM-based models of equal hidden size, even though they use fewer parameters and run substantially faster. Our experiments show that the speed and accuracy advantages remain consistent across tasks and at both word and character levels.

Extensions to both CNNs and RNNs are often directly applicable to the QRNN, while the model's hidden states are more interpretable than those of other recurrent architectures as its channels maintain their independence across timesteps. We believe that QRNNs can serve as a building block for long-sequence tasks that were previously impractical with traditional RNNs.

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

APPENDIX

BEAM SEARCH RANKING CRITERION

The modified log-probability ranking criterion we used in beam search for translation experiments is:

$$\log(P_{\text{cand}}) = \frac{T + \alpha}{T} \dots \frac{T_{\text{trg}} + \alpha}{T_{\text{trg}}} \sum_{i=1}^{T} \log(p(w_i|w_1 \dots w_{i-1})), \tag{9}$$

where $\alpha$ is a length normalization parameter (Wu et al., 2016), $w_i$ is the $i^{\text{th}}$ output character, and $T_{\text{trg}}$ is a "target length" equal to the source sentence length plus five characters. This reduces at $\alpha = 0$ to ordinary beam search with probabilities:

$$\log(P_{\text{cand}}) = \sum_{i=1}^{T} \log(p(w_i|w_1 \dots w_{i-1})), \tag{10}$$

and at $\alpha = 1$ to beam search with probabilities normalized by length (up to the target length):

$$\log(P_{\text{cand}}) \sim \frac{1}{T} \sum_{i=1}^{T} \log(p(w_i|w_1 \dots w_{i-1})). \tag{11}$$

Conveniently, this ranking criterion can be computed at intermediate beam-search timesteps, obviating the need to apply a separate reranking on complete hypotheses.

RESULTS ON COPY AND ADDITION TASKS

We take the addition task to mean sequence-to-sequence decimal addition of unaligned, variable-length numbers (the hardest of several versions). An example of this task for maximum length 5 digits is "73952+9462"→"83414". We train on $n_{\text{train}}$ randomly generated examples for up to 100 epochs with early stopping, and report the smallest model setup that achieves $> 99\%$ character-level validation accuracy, or the best validation accuracy achieved by any model setup if none achieve 99%.

For $n_{\text{digits}} = 5$ and $n_{\text{train}} = 100000$, the QRNN converges with models larger than 3 layers of 256 units, while in our experiments LSTMs require only 2 layers of 128 units. For $n_{\text{digits}} = 10$ and $n_{\text{train}} = 100000$, an LSTM reaches $98.5\%$ with 3 layers of 1024 units, while the best QRNN model (4 layers of 512 units) reaches only $95.0\%$.

The copy task is similarly implemented as sequence-to-sequence reconstruction of variable-length decimal numbers. For 5 digits, an example is "23532"→"23532". We train on 10000 randomly generated examples for up to 100 epochs. For $n_{\text{digits}} = 5$, the QRNN converges with models larger than 2 layers of 32 units or 1 layer of 256 units, while the LSTM requires only 1 layer of 32 units. For $n_{\text{digits}} = 10$, the QRNN requires a model larger than 2 layers of 128 units while the LSTM requires a model of at least 2 layers of 64 units or 1 layer of 256 units. For $n_{\text{digits}} = 40$, a QRNN with 5 layers of 512 units reaches $98.0\%$ while the best LSTM model, with 3 layers of 512 units, only reaches $95.5\%$.

A deeper LSTM would likely need some kind of residual or highway connections to converge on this task, while the deep QRNN converges relatively well despite our earlier experiences with 4-layer QRNNs without dense connections not converging successfully on the sentiment task.

