# Peer review of "Quasi-Recurrent Neural Networks"

_ICLR 2017 — accepted_

[Official Review · AnonReviewer2 · rating 7 · confidence 4 · 16 Dec 2016]
**Nice paper**
originality 2

This paper introduces a novel RNN architecture named QRNN.

QNNs are similar to gated RNN , however their gate and state update  functions depend only on the recent input values, it does not depend on the previous hidden state. The gate and state update functions are computed through a temporal convolution applied on the input.
Consequently, QRNN allows for more parallel computation since they have less  operations in their hidden-to-hidden transition depending on the previous hidden state compared to a GRU or LSTM. However, they possibly loose in expressiveness relatively to those models. For instance, it is not clear how such a model deals with long-term dependencies without having to stack up several QRNN layers.

Various extensions of QRNN, leveraging Zoneout, Densely-connected or seq2seq with attention, are also proposed.

Authors evaluate their approach on various tasks and datasets (sentiment classification, world-level language modelling and character level machine translation). 

Overall the paper is an enjoyable read and the proposed approach is interesting,
Pros:
- Address an important problem
- Nice empirical evaluation showing the benefit of their approach
- Demonstrate up to 16x speed-up relatively to a LSTM
Cons:
- Somewhat incremental novelty compared to (Balduzizi et al., 2016)

Few specific questions:
- Is densely layer necessary to obtain good result on the IMDB task. How does a simple 2-layer QRNN compare with 2-layer LSTM?  
- How does the i-fo-ifo pooling perform comparatively? 
- How does QRNN deal with long-term time depency? Did you try on it on simple toy task such as the copy or the adding task?

[Public Comment · (anonymous) · rating 5 · confidence 4 · 16 Dec 2016 (modified: 20 Jan 2017)]
**Strong, but should compare with PixelCNNs**

The authors describe the use of convolutional layers with intermediate pooling layers to more efficiently model long-range dependencies in sequential data compared with recurrent architectures. Whereas the use of convolutional layers is related to the PixelCNN architecture (Oord et al.), the main novelty is to combine them with gated pooling layers to integrate information from previous time steps. Additionally, the authors describe extensions based on zone-out regularization, densely connected layers, and an efficient attention mechanism for encoder-decoder models. The authors report a striking speed-up over RNNs by up to a factor of 16, while achieving similar or even higher performances.


Major comment
=============
QRNNs are closely related to PixelCNNs, which leverage masked dilated convolutional layers to speed-up computations. However, the authors cite ByteNet, which builds upon PixelCNN, only at the end of their manuscript and do not include it in the evaluation. The authors should cite PixelCNN already when introducing QRNN in the methods sections, and include it in the evaluation. At the very least, QRNN should be compared with ByteNet for language translation. How well does a fully convolutional model without intermediate pooling layers perform, i.e. what is the effect to the introduced pooling layers? Are their performance difference between f, fo, and ifo pooling? Did the authors investigate dilated convolutional layers?


Minor comments
==============
1. How does a model without dense connections perform, i.e. what is the effect of dense connections? To illustrate dense connections, the authors might draw them in figure 1 and refer to it in section 2.1. 

2. The run-time results shown in figure 4 are very helpful, but as far as I understood, the breakdown shown on the left side was measured for language modeling (referred in 3.2), whereas the dependency on batch- and sequence size shown on the right side for sentiment classification (referred in 3.1). I suggest to consistently show the results for either sentiment classification or language modeling, or both. At the very least, the figure caption should describe the task explicitly. Labeling the left and right figure by a) and b) would further improve readability. 

3. Section 3.1 describes a high speed-up for long sequences and small batch sizes. I suggest motivating why this is the case. While computations can be parallelized along the sequence length, it is less obvious why smaller batch sizes speed-up computations.

4. The proposed encoder-decoder attention is different from traditional attention in that attention vectors are not computed and used as input to the decoder sequentially, but on top of decoder output states in parallel. This should be described and motivated in the text.

Sentiment classification
------------------------
5. What was the size of the hold-out development set and how was it created? The text describes that data were split equally into training and test set, without describing the hold-out set.

6. What was the convolutional filter size?

7. What is the speed-up for the best hyper-parameters (batch size 24, sequence length 231)?

8. Figure 3 would be easier to interpret by actually showing the text on the y-axis. For the sake of space, one might use a smaller text passage, plot it along the x-axis, and show the activations of fewer neurons along the y-axis. Showing more examples in the appendix would make the authors’ claim that neurons are interpretable even more convincing.


Language modeling
-----------------
9. What was the size of the training, test, and validation set?

10. What was the convolutional filter size, denoted as ‘k’?

11. Is it correct that a very high learning rate of 1 was used for six epochs at the beginning?

12. The authors should show learning curves for a models with and without zone-out.
 
Translation
-----------
13. What was the size of the training, test, and validation set?

14. How does translation performance depend on k?

[Official Review · AnonReviewer3 · rating 7 · confidence 4 · 16 Dec 2016]
**good.**
soundness 3 · substance 2

This paper points out that you can take an LSTM and make the gates only a function of the last few inputs  - h_t = f(x_t, x_{t-1}, ...x_{t-T}) - instead of the standard - h_t = f(x_t, h_{t-1}) -, and that if you do so the networks can run faster and work better. You're moving compute from a serial stream to a parallel stream and also making the serial stream more parallel. Unfortunately, this simple, effective and interesting concept is somewhat obscured by confusing language.

- I would encourage the authors to improve the explanation of the model. 
- Another improvement might be to explicitly go over some of the big Oh calculations, or give an example of exactly where the speed improvements are coming from. 
- Otherwise the experiments seem adequate and I enjoyed this paper.

This could be a high value contribution and become a standard neural network component if it can be replicated and if it turns out to work reliably in multiple settings.

[Official Review · AnonReviewer1 · rating 6 · confidence 4 · 17 Dec 2016]
soundness 4 · originality 2 · impact 3 · meaningful comparison 1

This paper introduces the Quasi-Recurrent Neural Network (QRNN) that dramatically limits the computational burden of the temporal transitions in
sequence data. Briefly (and slightly inaccurately) model starts with the LSTM structure but removes all but the diagonal elements to the transition
matrices. It also generalizes the connections from lower layers to upper layers to general convolutions in time (the standard LSTM can be though of as a convolution with a receptive field of 1 time-step). 

As discussed by the authors, the model is related to a number of other recent modifications of RNNs, in particular ByteNet and strongly-typed RNNs (T-RNN). In light of these existing models, the novelty of the QRNN is somewhat diminished, however in my opinion their is still sufficient novelty to justify publication.

The authors present a reasonably solid set of empirical results that support the claims of the paper. It does indeed seem that this particular modification of the LSTM warrants attention from others. 

While I feel that the contribution is somewhat incremental, I recommend acceptance.

[Author Response · James Bradbury · 15 Jan 2017]
**Response to official reviews, plus results on copy and addition tasks**

Reviewer 1 asks, "Can you offer any results comparing the performance of the proposed Quasi-Recurrent Neural Network (QRNN) to that of the recently published ByteNet? How does it compare from a computational perspective?"

Yes. Nal Kalchbrenner released results for the ByteNet on the same IWSLT dataset we used in this paper in his slides for NIPS

[Final Decision · Program Chairs · 06 Feb 2017]
**ICLR committee final decision**

The paper is well written and easy to follow. It has strong connections to other convolutional models such as pixel cnn and bytenet that use convolutional only models with little or no recurrence. The method is shown to be significantly faster than using RNNs, while not losing out on the accuracy.
 
 Pros:
 - Fast model
 - Good results
 
 Cons:
 - Because of its strong relationship to other models, the novelty is incremental.